# Exploring Seminal Plasma GSTM3 as a Quality and In Vivo Fertility Biomarker in Pigs—Relationship with Sperm Morphology

**DOI:** 10.3390/antiox9080741

**Published:** 2020-08-12

**Authors:** Marc Llavanera, Ariadna Delgado-Bermúdez, Yentel Mateo-Otero, Lorena Padilla, Xavier Romeu, Jordi Roca, Isabel Barranco, Marc Yeste

**Affiliations:** 1Unit of Cell Biology, Department of Biology, Faculty of Sciences, University of Girona, ES-17003 Girona, Spain; marc.llavanera@udg.edu (M.L.); ariadna.delgado@udg.edu (A.D.-B.); yentel.mateo@udg.edu (Y.M.-O.); u1941676@campus.udg.edu (X.R.); 2Biotechnology of Animal and Human Reproduction (TechnoSperm), Institute of Food and Agricultural Technology, University of Girona, ES-17003 Girona, Spain; 3Department of Medicine and Animal Surgery, Veterinary Science, University of Murcia, ES-30100 Murcia, Spain; lorenaconcepcion.padilla@um.es (L.P.); roca@um.es (J.R.)

**Keywords:** pig, fertility, GSTM3, quality, seminal plasma

## Abstract

Glutathione S-transferases Mu 3 (GSTM3) is an essential antioxidant enzyme whose presence in sperm has recently been related to sperm cryotolerance, quality and fertility. However, its role in seminal plasma (SP) as a predictor of the same sperm parameters has never been investigated. Herein, cell biology and proteomic approaches were performed to explore the presence, origin and role of SP-GSTM3 as a sperm quality and in vivo fertility biomarker. GSTM3 in SP was quantified using a commercial Enzyme-Linked Immunosorbent Assay (ELISA) kit specific for *Sus scrofa*, whereas the presence of GSTM3 in testis, epididymis and accessory sex glands was assessed through immunoblotting analysis. Sperm quality and functionality parameters were evaluated in semen samples at 0 and 72 h of liquid-storage, whereas fertility parameters were recorded over a 12-months as farrowing rate and litter size. The presence and concentration of GSTM3 in SP was established for the first time in mammalian species, predominantly synthesized in the epididymis. The present study also evidenced a relationship between SP-GSTM3 and sperm morphology and suggested it is involved in epididymal maturation rather than in ejaculated sperm physiology. Finally, the data reported herein ruled out the role of this antioxidant enzyme as a quality and in vivo fertility biomarker of pig sperm.

## 1. Introduction

Artificial insemination (AI) is one of the major breakthroughs of pig reproductive biotechnology and has become the main technique for the breeding of this species worldwide, being an essential tool to achieve productivity challenges in swine industry [1]. Although AI can be performed using both frozen-thawed and liquid-stored sperm at 17 °C, the latter is used in the vast majority of pig AI [2,3]. While pig ejaculates are selected on the basis of sperm quality parameters prior to AI (i.e., motility, morphology and plasma membrane integrity), farrowing rates are often suboptimal after liquid-storage [3,4]. In this regard, it is estimated that about 6% of spermiogram-normal AI-pigs are subfertile individuals that remain “hidden,” which could lead to reproductive and economic losses [4,5]. Hence, exploring new sperm quality and fertility molecular biomarkers is essential to improve subfertility diagnosis and subsequent reproductive performance of pig AI-doses.

Pig seminal plasma (SP) is a complex mixture of secretions from testis, epididymis, seminal vesicles, bulbourethral glands and prostate that provides the physiological conditions for sperm during and after ejaculation [6]. As a result, SP is essential to preserve sperm metabolism and physiological status [7]. The complex composition of SP makes it likely to be a promising source of sperm quality and fertility biomarker candidates. Specifically, the protein fraction of SP has been found to be especially relevant for both sperm function and interaction with the female genital tract, even being essential for fertilization (reviewed from Reference [8]). Identification and quantification of differentially expressed proteins is known as comparative proteomics. The application of this emerging approach for the identification of novel SP quality and fertility biomarkers is currently flourishing [9]. In this regard, great efforts have been devoted to exploring new biomarkers in pig SP through comparative proteomics [10,11,12,13]. Specifically, recent studies uncovered the role of antioxidant enzymes in SP, such as glutathione peroxidase 5 (GPX5) and paraoxonase 1 (PON1), as sperm cryotolerance, quality and/or fertility biomarkers [14,15,16].

Glutathione S-transferases (GSTs) are essential antioxidant enzymes involved in cellular protection against oxidative stress, preserving sperm function and fertilizing ability (reviewed in Reference [17]). Antioxidant enzymes are known to be especially relevant for sperm cells since they are highly sensitive to oxidative stress due to the high amount of polyunsaturated fatty acids and their relatively low antioxidant capacity. Recent studies in pig and goat sperm evidenced the significance of GSTs in mammalian sperm physiology, highlighting their role on preserving mitochondrial function and maintaining plasma membrane stability [18,19]. Recently, the triple role of sperm GSTs has been well-established, being involved in cell detoxification, cellular signaling regulation and sperm-zona pellucida binding events [17]. In addition, previous research, including three recent meta-analysis, confirmed that some GSTs null-genotypes are a risk factor for male idiopathic subfertility or infertility in men [20,21,22,23,24,25,26,27]. Specifically, Aydemir et al. [28] reported that men with a specific GST Mu class null genotype showed increased oxidative stress in SP. Moreover, sperm GST Mu 3 (GSTM3) has been recently proposed as a quality [18], fertility [29] and cryotolerance [30] biomarker for pig sperm. Mounting evidence demonstrates the extracellular membrane-attached localization of sperm GSTs (reviewed from Reference [17]), surmising a potential direct relationship between sperm and SP-GSTM3. Therefore, SP-GSTM3 is a promising candidate to sperm quality and fertility biomarker.

In this regard, while sperm GSTM3 is well-characterized, the presence and putative function of seminal plasma GSTM3 (SP-GSTM3) on sperm function and fertility outcomes is yet to be investigated. Exploring the presence and role of pig SP-GSTM3 as a sperm quality and in vivo fertility biomarker is of utmost importance since it could improve and facilitate male subfertility diagnosis as well as give us some new insights into its molecular role as an antioxidant sperm enzyme.

## 2. Materials and Methods

### 2.1. Reagents

All chemicals were purchased from Sigma-Aldrich (St. Louis, MO, USA) unless otherwise stated. Fluorochromes were purchased from Molecular Probes (Thermo Fisher Scientific; Waltham, MA, USA). All reagents are listed in the Appendix A.

### 2.2. Animals and Samples

Semen and tissue samples were provided by an AI Spanish Centre (AIM Ibérica; Topigs Norsvin Spain SLU; Spain registration number (ES300130640127; August 2006) and European Union registration number (ES13RS04P; July 2012)). Production of the seminal AI-doses used in this study followed the current Spanish and European legislation for both commercialization of pig semen and animal health and welfare. Entire ejaculates were collected from 36 healthy and sexually mature (1–3 years-old) AI-pigs of four different breeds (Duroc (*n* = 8), Landrace (*n* = 13), Large White (*n* = 6) and Pietrain (*n* = 9)). A semi-automatic collection method was used (Collectis^®^, IMV Technologies, L’Aigle, France). Pigs were undergoing twice semen collections per week at the time of sample obtention.

### 2.3. Experimental Design

#### 2.3.1. Relationship Between SP-GSTM3 Concentration and Sperm Quality and Functionality Parameters

Entire ejaculates from 20 AI-boars (one ejaculate per boar) were collected and split into three aliquots. The first aliquot was used to assess sperm concentration and morphology. The second one was extended like an AI-dose (30 × 10^6^ sperm/mL in Biosem+; Magapor S.L., Ejea de los Caballeros, Spain) and used to evaluate sperm quality and functionality parameters immediately after ejaculate collection (0 h) and after 72 h of storage at 17 °C. The third aliquot was centrifuged twice at 1500× *g* for 10 min at room temperature (RT) to harvest SP. Thereafter, SP samples were examined under a microscope (Eclipse E400; Nikon, Tokyo, Japan) ensuring the absence of sperm. Finally, SP samples were immediately aliquoted (3 mL) and stored at −80 °C until thawed for SP-GSTM3 concentration analysis.

#### 2.3.2. Expression of GSTM3 in Boar Testis, Epididymis and Accessory Sexual Glands

In order to uncover the putative contribution of the testis, epididymis and accessory sexual glands to SP-GSTM3 content, a total of three healthy AI-boars were slaughtered (slaughterhouse La Mata de los Olmos, Teruel, Spain) for genetic replacement reasons. Genital tracts (medial testis; caput, corpus and cauda of the epididymis; mid-areas of the prostate; seminal vesicles; and bulbourethral glands) were dissected out to collect tissue samples (1 cm × 1 cm and 1 mm thick) and immediately frozen into liquid nitrogen and stored at −80 °C until Western blot analysis.

#### 2.3.3. Relationship Between SP-GSTM3 Content and In Vivo Fertility of Liquid-Stored Semen Samples

Three entire ejaculates from 16 AI-boars were collected over a 12-month period (one ejaculate every 4 months). These ejaculates were centrifuged for SP harvesting and the resulting SP-samples were stored at −80 °C for GSTM3 content analysis. For this 12-month period, weaned multiparous sows (1–7 litters produced) were cervically inseminated (2–3 times per estrus) using AI-doses (2400 × 10^6^ of total spermatozoa in 80 mL Biosem+) from these 16 AI-boars. Sows (Landrace and Large White) were housed in different Spanish farms and subjected to the same housing and management conditions. Fertility parameters, that is, farrowing rate (the proportion of inseminated sows that farrowed) and litter size (the total number of piglets born per litter), were recorded for this 12-month period. The number of inseminated sows was 3017 (more than 100 sows per boar).

### 2.4. Sperm Quality and Functionality Assessment

Sperm concentration, morphology, total and progressive motility, viability and acrosome integrity were assessed as quality parameters, whereas sperm membrane lipid disorder and intracellular hydrogen peroxide levels were assessed as functionality parameters.

Sperm concentration was measured automatically using a cell counter (NucleoCounter^®^ NC-100^TM^; ChemoMetec, Allerod, Denmark). Sperm morphology was assessed in semen samples extended (1:1; *v:v*) with 0.12% formaldehyde saline solution (Panreac, Barcelona, Spain). Sperm morphology evaluation was performed under a phase contrast microscope at 1000× magnification coupled with a SCA^®^ Production software (Sperm Class Analyzer Production, 2010; Microptic S.L., Barcelona, Spain). A total of 200 sperm per sample were evaluated and classified into the following categories: morphologically normal spermatozoa, acrosome abnormalities, folded and coiled tails, proximal and distal droplets and abnormal head size and shape. Sperm motility was assessed through a computer assisted sperm analyzer (CASA, ISASV1^®^, Proiser R+D S.L., Paterna, Spain). For this assessment, 5 μL of sperm at 20 × 10^6^ sperm/mL was loaded onto a pre-warmed (38 °C) Makler counting chamber (Sefi Medical Instruments, Haifa, Israel). Two replicates per sample, with a minimum of 600 sperm per each replicate, were assessed. The recorded sperm motility parameters were the percentage of motile sperm, with an average path velocity ≥20 μm/s and the percentage of sperm with progressive movement, showing a straight-line velocity ≥40 μm/s. The corresponding mean ± SEM was subsequently calculated.

Sperm viability, acrosome damage, membrane lipid disorder and intracellular hydrogen peroxide levels were assessed by flow cytometry. Sperm parameters were evaluated using a BD FACS Canto II flow cytometer (Becton Dickinson & Company, Franklin Lakes, NJ, USA) in semen samples extended at 30 × 10^6^ sperm per mL in Biosem+. Three technical replicates with a minimum of 10,000 sperm events positive to Hoechst 33342 (H-42) dye per sample were evaluated. Plasma membrane (viability) and acrosome integrities were assessed by triple-staining using H-42, propidium iodide (PI) and fluorescein-conjugated peanut agglutinin (PNA-FITC). Sperm samples (100 μL) were incubated with 3 μL of H-42 (0.5 mg/mL in PBS 1×), 2 μL of PI (0.5 mg/mL in PBS 1×) and 2 μL of PNA-FITC (100 µg/mL in PBS1×) for 10 min at 38 °C in the dark. Following this, sperm samples were diluted in 400 μL of PBS and subsequently analyzed through flow cytometry. Results were presented as the percentage of viable sperm (H-42^+^/PI^−^) with intact acrosome membrane (PNA-FITC^−^).

Sperm membrane lipid disorder was assessed by incubating semen samples (50 µL) with 2.5 μL of H-42 (0.05 mg/mL in PBS 1×) and 10 μL of Yo-Pro-1 (2.5 μM in dymetil sulfoxide (DMSO)) for 8 min at 38 °C in the dark. Next, 26 μL of Merocyanine 540 (M-540, 0.1 mM in DMSO) was added to each sample prior to incubation for 2 min at 38 °C in the dark. Results were presented as the percentage of viable sperm (H-42^+^/Yo-Pro-1^−^) with high plasma membrane lipid disorder (M-540^+^). Finally, intracellular hydrogen peroxide levels were evaluated through the incubation of sperm samples (50 µL) with 1.5 μL of H-42 (0.05 mg/mL in PBS 1×), 1 μL of PI (0.05 mg/mL in PBS 1×) and 1 μL of 5- and 6-chloromethyl-2, 7-dichlorodihydrofluorescein diacetate acetyl ester (CM-H_2_DCFDA; 1 mM in DMSO) in 950 μL of PBS for 30 min at 38 °C in the dark. A sample of each semen samples was incubated with 1 μL of tert-butyl hydroperoxide solution (70% in distilled water) and used as a positive control. Results are presented as the percentage of viable sperm (H-42^+^/Yo-Pro-1^−^) with high intracellular hydrogen peroxide levels (DCF^+^).

### 2.5. Western Blot Analysis

Tissue samples from testis and accessory glands were lysed through a hybrid method combining both chemical and mechanical lysis. A total of 50 mg of tissue was resuspended in 800 μL of lysis buffer (xTractor™ Buffer; Takara Bio, Mountain View, CA, USA) supplemented with 50 U DNase I (Takara Bio), 1% protease inhibitor cocktail and sodium orthovanadate (700 mM). Samples were vortexed and incubated for 10 min at 4 °C. Subsequently, samples were disrupted mechanically four times through a TissueLyser II (Qiagen, Hilden, Germany) set at 30 strokes/s for 5 min at 4 °C. Finally, samples were centrifuged at 12,000× g for 30 min at 4 °C and supernatants were stored at −80 °C prior to total protein quantification. Quantification of total protein was carried out in triplicate by a detergent compatible (DC) method (BioRad). Fifteen micrograms of total protein were resuspended in Laemmli reducer buffer 2× (BioRad) and heated at 95 °C for 7 min. Subsequently, samples were loaded onto a gradient (8–16%) polyacrylamide gel (Mini-PROTEAN^®^ TGX Stain-FreeTM Precast Gels, Bio-Rad) and electrophoresed at 150 V for 90 min. Next, total protein was visualized by UV exposition and acquisition using a G:BOX Chemi XL system (SynGene, Frederick, MD, USA). Following this, proteins from gels were transferred onto polyvinylidene fluoride (PVDF) membranes using Trans-Blot^®^ TurboTM (Bio-Rad), which were subsequently blocked in blocking buffer (10 mmol/L Tris, 150 mmol/L NaCl and 0.05% Tween-20; pH = 7.3 and 5% bovine serum albumin (Roche Diagnostics, S.L., Basel, Switzerland) for 1 h at room temperature (RT). Blocked membranes were incubated with primary anti-GSTM3 antibody (ref. ARP53561_P050; 0.05 μg/mL) for 1 h with agitation at RT. Subsequently, membranes were washed thrice with TBS1×-Tween20 (10 mmol/L Tris, 150 mmol/L NaCl and 0.05% Tween-20; pH = 7.3) and incubated with a secondary anti-rabbit antibody conjugated with HRP for 1 h with agitation (ref. P0448; 0.025 μg/mL) at RT. Finally, membranes were washed five times and visualized with a chemiluminescence substrate (Immobilion^TM^ Western Detection Reagents, Millipore) prior to be scanned with G:BOX Chemi XL 1.4. A peptide competition assays utilizing 20-fold GSTM3 immunizing peptide with regard to the antibody was performed to confirm the specificity of the GSTM3-primary antibody. Three replicates per sample were evaluated.

### 2.6. Enzyme-Linked Immunosorbent Assay (ELISA)

Porcine GSTM3 in SP-samples was quantified using a porcine-specific competitive ELISA kit (MBS7260929; MyBioSource, San Diego, CA, USA) following the manufacturer’s guidelines. In brief, 100 μL of GSTM3 standards (0, 10, 25, 50, 100, 250 ng of GSTM3/mL) were loaded onto the corresponding wells to obtain the standard curve. The same volume of SP was loaded onto their corresponding wells. Then, samples and standards were incubated together with 50 μL of HRP-conjugated GSTM3 for 1 h at 37 °C. Subsequently, wells were washed five times and incubated with the substrate of HRP enzyme for 15 min at 37 °C. Finally, stop solution was added and the color intensity was measured spectrophotometrically at 450 nm in a microplate spectrophotometer (BioTek Epoch; BioTek, Winooski, Vermont, USA). A standard curve relating the absorbance (ABS) to the GSTM3 concentration of standards was plotted. The logarithmic regression curve was subsequently calculated and the GSTM3 concentration (GSTM3) of each sample was interpolated from the following standard curve:(1)$$\left[\mathrm{GSTM}3\right]=-109.5\ln\left(\mathrm{ABS}\right)+95,587\;R^{2}=0.962.$$

The ELISA kit was highly specific for porcine GSTM3, with a sensitivity of 1.0 ng/mL and a detection range of 94–103%. The blank control wells contained PBS 1× (pH = 7.0). All standards and samples were loaded in duplicate.

### 2.7. Statistical Analysis

Data were evaluated using a statistical package (IBM SPSS Statistics 25.0; IBM Corp., Armonk, NY, USA). First of all, normal distribution and homogeneity of variances were tested through Shapiro-Wilk and Levene tests, respectively.

Differences of GSTM3 concentration in SP (SP-GSTM3) between breeds were tested through one-way analysis of variance (ANOVA; factor: breed; independent variable: SP-GSTM3) followed by post-hoc Sidak for pair-wise comparisons. Correlations between SP-GSTM3 and sperm quality and functionality parameters evaluated upon ejaculation (0 h) were determined through Pearson coefficient. Correlations between SP-GSTM3 and the variations within total and progressive sperm motility, viability, acrosome damage, intracellular hydrogen peroxide levels and membrane stability throughout liquid-storage at 17 °C (i.e., 0 vs. 72 h) were also calculated through Pearson correlation coefficient.

Boar reproductive performance data were corrected for parameters related to farm and sow through a multivariate statistical model, as described in Broekhuijse et al. [31]. The resulting deviations in fertility parameters (farrowing rate deviation, FR; and litter size deviation, LS) were used to classify the 16-AI boars into two groups (high FR and LS; low FR and LS). This classification was conducted through a hierarchical cluster analysis based on the nearest neighbor approach and the squared Euclidean distance (SED). Following this, the SP-GSTM3 concentration evaluated with ELISA was compared between the two fertility groups (i.e., high and low FR/LS) through a *t*-test for independent samples.

The level of significance was set at *p* < 0.05 and data are shown as mean ± standard error of the mean (SEM).

## 3. Results

### 3.1. Characterisation of Porcine SP-GSTM3

The GSTM3 concentration in SP from pig ejaculates was 61.62 ± 2.18 ng/mL, ranging from 38.26 to 81.82 ng/mL. No differences in SP-GSTM3 concentration levels were found between breeds (*p* > 0.05): Duroc (60.64 ± 4.34 ng/mL), Landrace (60.24 ± 3.47 ng/mL) and Pietrain (64.49 ± 3.87 ng/mL) (Figure 1).

### 3.2. Correlation between SP-GSTM3 and Sperm Quality and Functionality Parameters of Semen Samples

Sperm quality and functionality parameters (mean ± SEM and range) of semen samples assessed immediately after ejaculation (0 h) are shown in Table 1.

Figure 2 shows Pearson correlation coefficients between SP-GSTM3 concentration and sperm quality and functionality parameters of semen samples immediately after ejaculation (0 h). No correlation between SP-GSTM3 concentration and sperm quality and functionality parameters were found (*p* > 0.05), except for morphology parameters. The SP-GSTM3 concentration was positively correlated with the percentage of sperm with normal morphology (R = 0.501; *p* < 0.05) and negatively correlated with the percentages of sperm with proximal droplets (R = −0.454; *p* < 0.05), distal droplets (R = −0.604; *p* < 0.05) and coiled tails (R = −0.574; *p* < 0.05).

### 3.3. Relationship between SP-GSTM3 Concentration and Sperm Resilience to Withstand Liquid-Storage at 17 °C

Total and progressive sperm motility, viability, acrosome damage, intracellular hydrogen peroxide levels and membrane stability were assessed at 0 and 72 h of liquid-storage. The difference of the percentage in each sperm parameter between both evaluation time-points (0 and 72 h) was calculated to evaluate the putative relationship between the ability of semen samples to withstand liquid-storage at 17 °C and SP-GSTM3 concentration. Subsequently, the decline of each parameter between 0 and 72 h and SP-GSTM3 concentrations were compared through Pearson correlations (Figure 3). No significant correlations between SP-GSTM3 concentration and the difference in percentages of each sperm quality and functionality parameters between both evaluation time-points were found (*p* > 0.05).

### 3.4. Presence of GSTM3 in SP-Related Testis and Accessory Sexual Glands

Immunoblotting analysis were performed to elucidate the putative contribution of testis, epididymis and accessory glands on GSTM3 content in SP. As shown in Figure 4, immunoblotting of GSTM3 reported a single or double-band pattern of ~25 and ~75 kDa, depending on the tissue type. Specifically, the testis (T) and caput epididymis (HE) showed both ~25 and ~75 kDa-bands, whereas the corpus (BE) and caput (TE) of the epididymis, the prostate (P) and seminal vesicles (SV) showed a single ~75 kDa-band. Peptide competition assay confirmed the GSTM3-specificity of all bands. Remarkably, GSTM3 band-signal intensity was found to be higher in the caput epididymis than in other accessory sexual glands. However, no GSTM3 signal was found in bulbourethral glands (B).

### 3.5. Relationship between SP-GSTM3 and In Vivo Fertility Outcomes

A total of 16-AI boars was classified through hierarchical clustering (*p* < 0.001) into two groups according to their farrowing rate and litter size deviation (low fertility and high fertility boars). The six AI-boars that exhibited the highest (*n* = 3) and lowest (*n* = 3) farrowing rate (FR) and litter size (LS) deviation were selected for SP-GSTM3 analysis (Figure 5A). A dot plot of FR and LS deviation showing the selected individuals from the 16 AI-boars is shown in Appendix A. Farrowing rate and litter size deviation significantly differed between fertility groups (*p* < 0.05; Figure 5A). The concentration of GSTM3 was assessed in SP-samples from three ejaculates of each boar. No differences were found when comparing SP-GSTM3 concentrations between SP-samples from low (64.66 ± 6.52 ng/mL) and high (64.66 ± 7.99 ng/mL) fertility groups (*p* > 0.05) (Figure 5B).

## 4. Discussion

Given the role of sperm GSTM3 as a sperm quality [18], fertility [29] and cryotolerance [30] biomarker and its extracellular membrane-attached localization in mammalian species (reviewed in Reference [17]), SP-GSTM3 is likely to be related to those sperm parameters. Therefore, it is reasonable to suggest that exploring SP-GSTM3 as a sperm quality and in vivo fertility biomarker may improve the evaluation of reproductive performance of pig AI-doses. To the best of our knowledge, this is the first report confirming the presence of GSTM3 in SP of mammals, which underpins the contribution of testis, epididymis and accessory sexual glands to GSTM3 content in SP and assessing the putative role of SP-GSTM3 as a molecular biomarker.

The results of the present study confirmed the presence and concentration of GSTM3 in SP for the first time in any species. The average SP-GSTM3 concentration was 61.62 ± 2.18 ng/mL, ranging from 38.26 to 81.82 ng/mL. As far as we are aware, no information regarding GSTs concentration in SP has been reported in the literature. However, the mean concentration of GSTM3 in pig SP was higher from other antioxidant enzymes such as glutathione peroxidase 5 [14] (GPX5; 9.63–30.13 ng/mL) and paraoxonase 1 [32] (PON1; 0.96–1.67 ng/mL). Another objective of the present study was to compare SP-GSTM3 levels between pig breeds. Although differences in ejaculate volume, sperm concentration and percentage of viable sperm have been extensively reported across pig breeds [33], our results did not show differences in the SP-GSTM3 content between Duroc, Landrace and Pietrain breeds. In this context, it is worth bearing in mind that the class-clustered organization of GST genes in both plant and animals reveals their importance during evolutionary history. Furthermore, GSTs are ubiquitous and highly conserved enzymes among species (reviewed from Reference [17]). Therefore, the fact that GSTs are highly conserved proteins would support the similar GSTM3 concentrations found in the SP of these three pig breeds.

The putative relationship between SP-GSTM3 concentration and sperm quality and functionality parameters was measured using Pearson correlation coefficients. While sperm GSTM3 has been proposed as a quality [18] and cryotolerance [30] biomarker, the role of SP-GSTM3 as a predictor of quality and functionality of ejaculated sperm had never been explored. Interestingly, in the present study, no correlations of SP-GSTM3 with ejaculate volume, sperm concentration, motility, viability, acrosome damage, membrane lipid disorder and ROS levels were found. However, a clear relationship between SP-GSTM3 and sperm morphology was observed. Specifically, higher concentrations of GSTM3 in SP were significantly associated to a lower percentage of sperm exhibiting proximal and distal droplets and coiled tails. Concomitantly, higher SP-GSTM3 levels were related to a higher percentage of sperm with normal morphology. It is widely known that sperm malformations could have their origin in the testis (primary malformations) or in the epididymis (secondary malformations) [34,35]. All sperm abnormal morphologies related to SP-GSTM3 (proximal and distal droplets and coiled tails) are categorized as secondary malformations and therefore are a result of an inadequate or poor epididymal maturation. Cytoplasmatic droplets are originated in the testis and move distally during epididymal maturation [35,36]. Both distal and proximal droplets are considered as sperm malformations since they have been related to male infertility in domestic species and indicate the failure of epididymal maturation (reviewed from References [35,36]). On the other hand, coiled tails are formed during sperm epididymal maturation, probably because of the weakness of dense fibers [37]. Other studies in men showed significant correlations between sperm morphology and the content and/or activity of some antioxidant enzymes in SP such as superoxide dismutase (SOD), catalase (CAT) and GPX [38,39]. Against this background, it is suggested that GSTM3 in SP plays a key role during epididymal maturation and is proposed as a sperm morphology biomarker candidate. However, further research regarding its molecular role upon sperm epididymal maturation is required to confirm this hypothesis.

Immunoblotting analyses of the testis, epididymis and accessory glands were performed to elucidate their contribution to SP-GSTM3 secretion. The presence of GSTM3 was confirmed by a double-band pattern of ~25 and ~75 kDa in the testis and cauda epididymis and a single band of ~75 kDa in the corpus and cauda epididymis, the prostate and seminal vesicles. No GSTM3-signal was found in bulbourethral glands. Previous studies in pigs reported a single band of ~25 kDa in sperm samples [29,30], which corresponds to its molecular mass. The GSTM3-specific ~75 kDa-band reported herein in tissue samples could be attributed to either GSTM3 homo- or hetero-trimerization; however, further research to confirm this hypothesis is much warranted. Expression of GSTM3 was found to be higher in the epididymis than the testis and accessory glands. As aforementioned, SP is a mixture of secretions from the testis, epididymis and accessory sexual glands. In this regard, the testis, epididymis and accessory glands, except for bulbourethral glands, contribute to GSTM3 content in SP. Moreover, the fact that SP-GSTM3 is mainly synthesized in the epididymis is an evidence that would support the role of this enzyme during epididymal maturation and the occurrence of secondary sperm morphology abnormalities. Accordingly and based on the results of the present study, poor synthesis of GSTM3 in the epididymis could lead to an inadequate epididymal maturation of sperm, which could be detected in SP.

The ability of SP-GSTM3 of predicting sperm resilience to withstand liquid-storage at 17 °C was assessed for the first time in any mammalian species. Considering that other antioxidant enzymes in SP such as GPX5 or SOD have shown to be quality predictors of AI-doses during liquid-storage [14,40], GSTM3 would also be expected to be a good biomarker. Although recent reports showed sperm-GSTM3 as a biomarker of sperm resilience to withstand liquid-storage and cryopreservation [18,30], our findings did not find SP-GSTM3 to be a good predictor. In effect, the results reported herein did not show significant correlations between SP-GSTM3 concentrations and the decline in sperm quality and functionality parameters during liquid-storage of semen at 17 °C. Different studies in goats and pigs showed the importance of sperm membrane-attached GSTM3 for mitochondrial function, plasma membrane stability and oxidative regulation [18,19], thus evidencing its molecular role in sperm physiology. The lack of correlation between SP-GSTM3 content and the sperm resilience to withstand liquid-storage would indicate the absence of molecular effects of this antioxidant enzyme upon sperm physiology. It is hypothesized that the presence of GSTM3 in SP could correspond to the remaining content of its activity in the epididymis during sperm maturation, rather than being physiological active upon ejaculated sperm.

Finally, the role of SP-GSTM3 to be an in vivo fertility biomarker was explored. The relevance of antioxidant enzymes from SP as fertility biomarkers is not clear, since it has been found to differ between molecular types. Recent studies showed the importance of SP-GPX5 as a relevant fertility biomarker of pig semen [14], whereas SOD turned out not to be related to sperm fertilizing ability [40]. Although sperm GSTM3 was stablished as an in vivo fertility biomarker in pigs [29], the results of the present study did not show any effect of SP-GSTM3 concentration upon in vivo fertility outcomes of AI-boars. However, the ejaculates used here were obtained from an AI-center, which selects their boars on the basis of their reproductive performance. Therefore, the good fertility of the boars used in this study could mask the real effects of SP-GSTM3 on its fertility. Conducting similar experiments using non-selected species, such as humans, is recommended to confirm our results in other mammalian species.

## 5. Conclusions

In conclusion, the data reported in the present study established the presence and concentration of GSTM3 in pig SP, remaining similar between boar breeds. On the other hand, SP-GSTM3 was reported to be predominantly synthesized in the epididymis and its concentration was found to be negatively correlated to abnormal sperm morphology. Indeed, low GSTM3 content in SP, mainly synthesized during sperm transport through the epididymis, was found to be related to increased percentage of secondary sperm malformations (coiled tails and proximal and distal droplets). Moreover, a lack of correlation between SP-GSTM3 content and the resilience of sperm to withstand liquid-storage was also observed. While SP-GSTM3 is thus suggested to have a molecular role during epididymal maturation rather than being involved in the physiology of ejaculated sperm, further studies using GSTs inhibitors are required in order to confirm this hypothesis. Finally, whilst the findings of the present study supported the use of SP-GSTM3 as a good sperm morphology predictor, they ruled out its relationship with other sperm quality parameters or with boar reproductive performance.

## Figures and Tables

**Figure 1 antioxidants-09-00741-f001:**
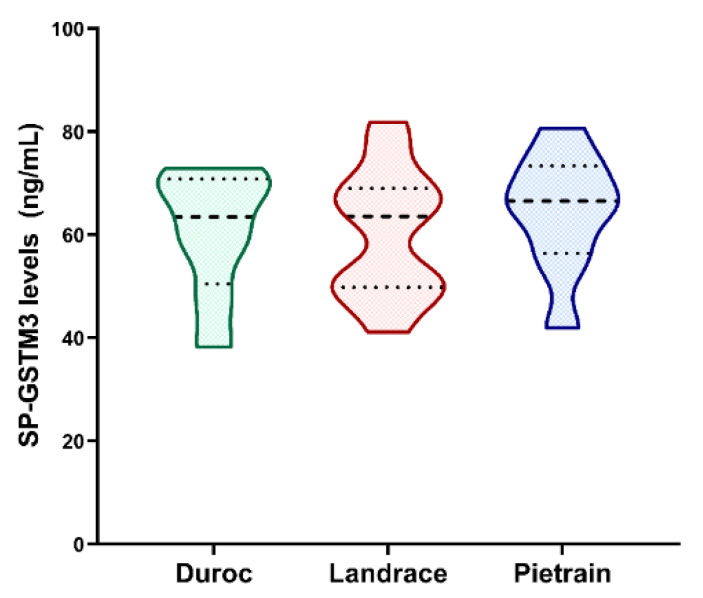
Violin plots showing seminal plasma GSTM3 (SP-GSTM3) concentration levels (ng/mL) and distribution from ejaculates of different pig breeds. Duroc, *n* = 8; Landrace, *n* = 13 and Pietrain, *n* = 9. Dashed line represents the median and dotted lines the 25 and 75% quartiles. No significant differences (*p* > 0.05) in SP-GSTM3 concentrations were found between breeds.

**Figure 2 antioxidants-09-00741-f002:**
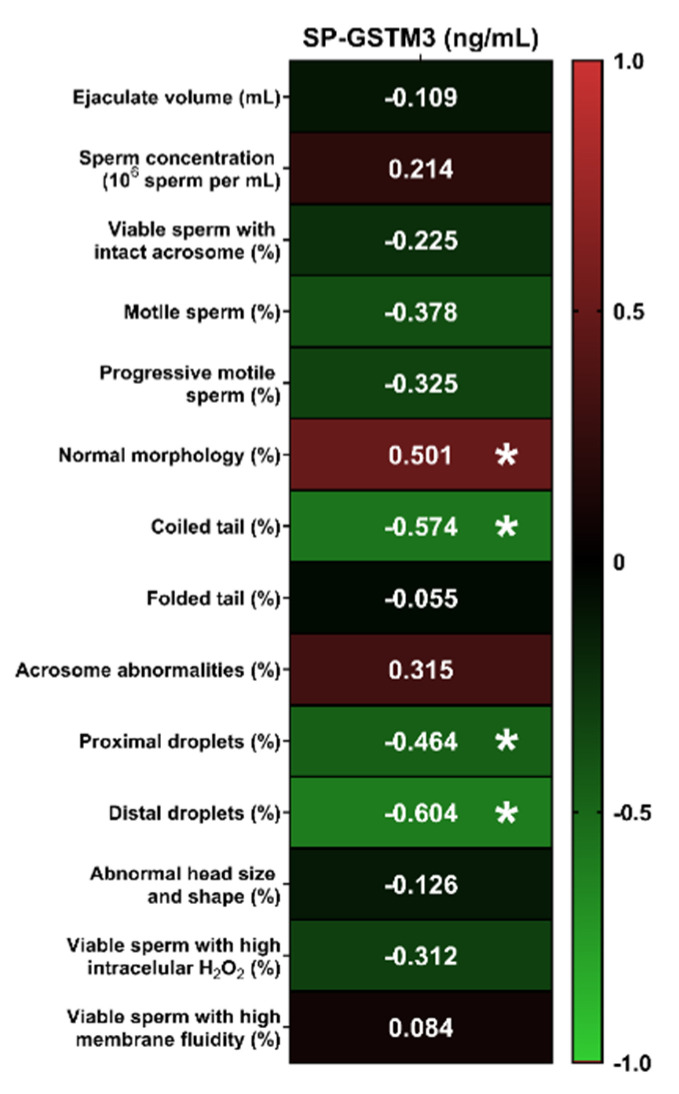
Heatmap of Pearson correlation coefficients between pig seminal plasma GSTM3 (SP-GSTM3) concentrations (ng/mL) and sperm quality and functionality parameters assessed in semen samples immediately after ejaculation (*n* = 20). * *p* < 0.05.

**Figure 3 antioxidants-09-00741-f003:**
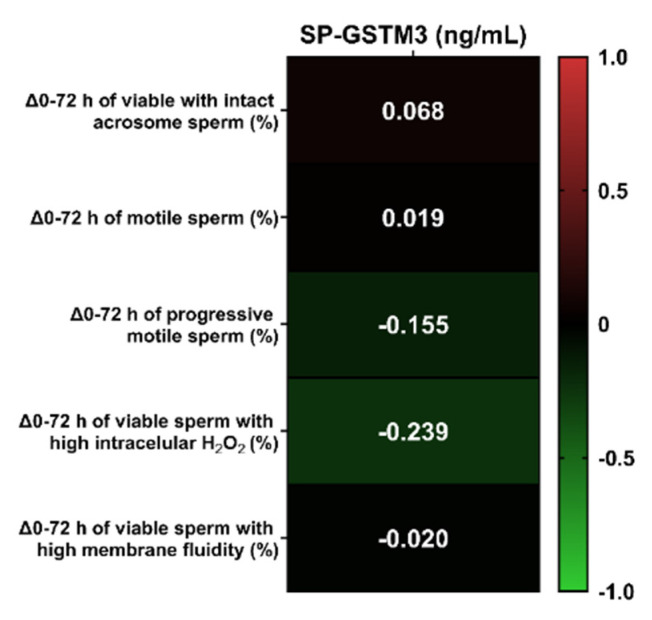
Heatmap of Pearson correlation coefficients between seminal plasma GSTM3 (SP-GSTM3) concentrations (ng/mL) and the differences of sperm quality and functionality parameters between 0 and 72 h of storage (*n* = 20). No significant correlations were found (*p* > 0.05).

**Figure 4 antioxidants-09-00741-f004:**
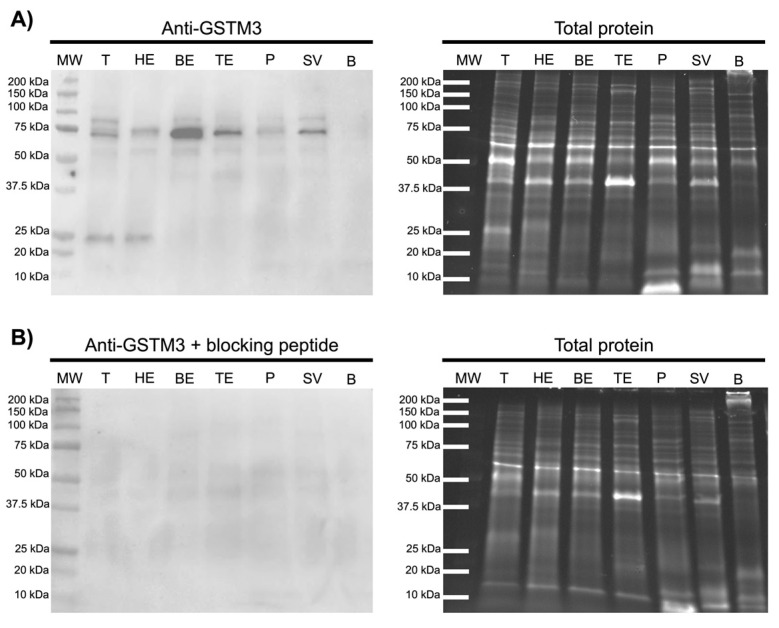
Representative Western blot resulting from incubation with the (**A**) GSTM3 antibody (Anti-GSTM3), (**B**) its corresponding peptide competition assay (Anti-GSTM3 + blocking peptide) and their loading controls (Total protein). MW: molecular weight. T: testis. HE: caput epididymis. BE: corpus epididymis. TE: cauda epididymis. P: prostate. SV: seminal vesicles. B: bulbourethral glands.

**Figure 5 antioxidants-09-00741-f005:**
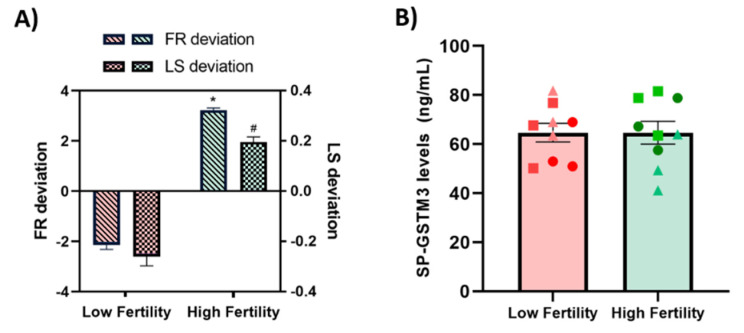
(**A**) Mean ± standard error of the mean (SEM) of farrowing rate (FR; lines) and litter size (LS; squares) deviation of samples classified as low fertility (red) and high fertility (green) boars. The six Artificial Insemination (AI)-boars were classified as having low (*n* = 3) or high (*n* = 3) fertility outcomes deviation (FR and LS). Different symbols (*, #) indicate significant differences (*p* < 0.05) between fertility groups. (**B**) Mean ± SEM of the concentration (ng/mL) of GSTM3 in seminal plasma (SP-GSTM3) were assessed in three ejaculates per boar, categorized as low (red) and high (green) fertility boars. Different symbols (●, ■, ▲) represent different ejaculates from the same boar within a fertility group. No significant differences in SP-GSTM3 content were found between fertility groups.

**Table 1 antioxidants-09-00741-t001:** Sperm quality and functionality parameters of pig semen samples assessed immediately after ejaculation (*n* = 20).

Sperm Quality and Functionality Parameters	Mean ± SEM	Range (Min–Max)
Ejaculate volume (mL)	619.05 ± 21.28	357–729
Sperm concentration (106 sperm per mL)	171.93 ± 10.85	91.65–256
Viable sperm with intact acrosome (%)	84.73 ± 1.47	72.10–91.60
Motile sperm (%)	76.85 ± 2.03	51–90
Progressive motile sperm (%)	50 ± 2.36	26–66
Normal morphology (%)	77.95 ± 3.14	40–95
Coiled tails (%)	0.30 ± 0.13	0–2
Folded tails (%)	6.25 ± 1.26	0–19
Acrosome abnormalities (%)	3.32 ± 0.95	0–17
Proximal droplets (%)	6 ± 1.55	0–26
Distal droplets (%)	5.45 ± 1.57	0–29
Abnormal head size and shape (%)	0.90 ± 0.35	0–5
Viable sperm with high intracellular H2O2 (%)	30.69 ± 3.58	3.40–56.40
Viable sperm with high plasma membrane fluidity (%)	1.69 ± 0.19	0.50–3.50

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
