# Peer review of "Exploring Seminal Plasma GSTM3 as a Quality and In Vivo Fertility Biomarker in Pigs—Relationship with Sperm Morphology"

_antioxidants, 2020, doi:10.3390/antiox9080741_

Round 1

Reviewer 1 Report

  1. In the introduction the functions of GSTM3 should be described more exactly (if such information is available).
  2. The list of abbreviation used in the text added at the beginning of the article would increase readability of the manuscript.
  3. whether the experiment required the approval of ethical commission? If Yes – please add the name of committee and number of agreement. If not – the Authors should give this fact in the text.
  4. The list of reagents in point 2.1. Reagents would increase the readability of methods described in the manuscript.
  5. The Reviewer known that the number of pigs are mentioned in further parts of “Material and methods” but in points 2.2. “Animals and samples” the number of pigs (within each breed) included into the experiment and their age should be added (maybe tabular form would be appropriate?)
  6. the reference list should be checked. In some references the year of publication is written in bolt, in other not in bold

In my opinion the article may be published after minor revisions

Author Response

Reviewer #1

Comment 1: In the introduction the functions of GSTM3 should be described more exactly (if such information is available).

Answer: Thank for your valuable suggestion. Accordingly, we have extended the description of the GSTM3 functions in the Introduction section (lines 61-64).

Comment 2: The list of abbreviation used in the text added at the beginning of the article would increase readability of the manuscript.

Answer: Thank you for your comment. We inquired the Editorial Office of Antioxidants about the possibility of adding a list of abbreviations before the Introduction section. Unfortunately, this is not allowed as per the Journal guidelines. Instead, following your suggestions and Journal guidelines, we have included an abbreviation list at the end of the Manuscript.

Comment 3: Whether the experiment required the approval of ethical commission? If Yes – please add the name of committee and number of agreements. If not, the Authors should give this fact in the text.

Answer: Thanks for your question. The authors did not manipulate any animal, but rather all ejaculates were provided by an authorised AI-centre that routinely collected semen samples for commercial purposes. As indicated in the Manuscript (lines 83-87), this centre complies with Spanish (ES300130640127, August 2006) and European (ES13RS04P, July 2012) guidelines for both animal health and welfare, collection of ejaculates, and commercialization of AI-doses. Moreover, all animals used in the present study were sent to the abattoir when it was considered that they had to be culled. The abattoir provided us with the tissue samples used in this study. Therefore, the authors did not manipulate any animal nor was the animal manipulation conducted for a research purpose. Thus, in our opinion, there is no ethical concern; however, we provided the authorization code for the commercial stud centres, since this demonstrates that the work fulfilled with the guidelines of Animal health and welfare.

Comment 4: The list of reagents in point 2.1. Reagents would increase the readability of methods described in the manuscript.

Answer: We appreciate your comment. According to your suggestion, we listed all the reagents used in this study, including their corresponding references, in a file added to Supplementary Materials (Supplementary File 1).

Comment 5: The Reviewer known that the number of pigs are mentioned in further parts of “Material and methods” but in points 2.2. “Animals and samples” the number of pigs (within each breed) included into the experiment and their age should be added (maybe tabular form would be appropriate?)

Answer: Thank you for your comment. Following your suggestions, we have included this information within the sub-section “2.2. Animals and samples” of the Material and Methods section (lines 87-89).

Comment 6: The reference list should be checked. In some references the year of publication is written in bolt, in other not in bold

Answer: Sorry for these typo mistakes. All references have been checked and corrected.

Reviewer 2 Report

The manuscript titled “Exploring seminal plasma GSTM3 as a quality and in 2 vivo fertility biomarker in pigs: relationship with 3 sperm morphology” is well written especially the discussion section. Herein, the author explores the role of SP GSTM3 as a fertility biomarker and established its role as a good sperm morphology predictor.

Few points that need to be addressed:

  1. Is there any correlation between ROS markers and GSTM3 expression and sperm morphology? Did the author use any ROS quencher in the study to establish the same?
  2. Is there a relationship between GSTM3 substrate and sperm properties?
  3. There are no experiments done to confirm the relationship between sperm GSTM3 and intracellular ROS regulation? More experiments are needed to establish the role of GSTM3 as a fertility biomarker. Use of other GST inhibitors is recommended for the same.

Author Response

Reviewer #2

Comment 1: Is there any correlation between ROS markers and GSTM3 expression and sperm morphology? Did the author use any ROS quencher in the study to establish the same?

Answer: Thank you very much for your comments. As shown in Figure 2, no correlation between concentration of GSTM3 in seminal plasma and ROS levels in viable sperm was found (intracellular H2O2 generation assessed with CM-H2DCFDA). Moreover, previous studies of our laboratory also found no correlation between ROS levels in viable spermatozoa and relative content of GSTM3 in sperm (Llavanera, et al., 2019). Interestingly, although GSTM3 is considered as an antioxidant enzyme, there is no information in the literature showing any correlation between the levels of GSTM3 in sperm or seminal plasma and sperm ROS levels in liquid-stored pig sperm. Regarding the second question, as the present study is highly focused on exploring the amounts of GSTM3 in seminal plasma as a quality, functionality and fertility biomarker of pig semen, we believe that performing manipulative experiments such as the use of ROS quenchers would make the results and discussion more complicated and confusing. That being said, we agree with the reviewer that the lack of correlation between this antioxidant enzyme and ROS levels is worth of investigation in that further studies focused on the molecular role of GSTM3 in sperm.

Comment 2: Is there a relationship between GSTM3 substrate and sperm properties?

Answer: Thanks for your question. To the best of our knowledge, there are no studies in the literature aimed at evaluating the relationship between endogenous glutathione (GSH) levels and quality and functionality parameters of pig sperm. Instead, some studies exploring the effects of supplementing pig semen extenders with GSH supplementation have evidenced a positive effect of this molecule on sperm quality during liquid storage at 17 °C (Zhang et al., 2016). However, as our study is focused on the predictive value of seminal plasma GSTM3 as a quality, functionality and fertility biomarker, we considered manipulative studies out of our goal.

Comment 3: There are no experiments done to confirm the relationship between sperm GSTM3 and intracellular ROS regulation? More experiments are needed to establish the role of GSTM3 as a fertility biomarker. Use of other GST inhibitors is recommended for the same.

Answer: Thank you for your comment. As indicated in the results section, no relationship between sperm ROS regulation and concentration of GSTM3 in seminal plasma was found (lines 247-249). The assessment of intracellular ROS levels was carried out following the well-established procedure described by Guthrie and Welch (2006). In addition, our previous study also demonstrated that sperm GSTM3 content was not related to intracellular ROS levels (Llavanera et al., 2019). Therefore, in our opinion, this relationship has already been studied. Regarding your valuable suggestion of further experiments to establish the role of GSTM3 as a fertility biomarker, we agree that the use of GSTs inhibitors would be very interesting to study the molecular function of this protein in fertilization process. However, two major limitations would come up: (i) the absence of a specific GSTM3 inhibitor, and (ii) the impossibility of specifically inhibit seminal plasma GSTM3 but not sperm GSTM3. Although we appreciate the reviewer's suggestion, our goal was to explore whether the amounts of GSTM3 in seminal plasma could be considered as a fertility biomarker for AI-boars in a real situation, considering that 99% of sows are artificially inseminated using liquid-stored semen at 17 ºC (Yeste et al., 2017). Under this scenario, we consider that our experimental design, which involves 16 AI-boars with 3,017 inseminated sows using liquid-stored AI-doses, is robust and confident enough to explore the role of seminal plasma GSTM3 as an in vivo fertility biomarker. While we keep in mind that the role of seminal plasma GSTM3 as an in vivo fertility biomarker could be masked by the high fertility records of the AI-boars, this shortcoming is discussed appropriately in the Discussion section (lines 389-393). Finally, we also agree with the reviewer that exploring the molecular role of seminal plasma GSTs, using GST inhibitors (especially in the case of sperm epididymal maturation), would be of great interest for further studies. Therefore, following the reviewer’s suggestion, an appropriate sentence has been included in the Discussion section (lines 402-404).

References

Guthrie, H.D., Welch GR. (2006). Determination of intracellular reactive oxygen species and high mitochondrial membrane potential in Percoll-treated viable boar sperm using fluorescence-activated flow cytometry. Journal of Animal Science, 84:2089–100.

Llavanera, M.; Delgado-Bermúdez, A.; Olives, S.; Mateo-Otero, Y.; Recuero, S.; Bonet, S.; Fernández-Fuertes, B.; Yeste, M.; Barranco, I. (2020). Glutathione S-Transferases Play a Crucial Role in Mitochondrial Function, Plasma Membrane Stability and Oxidative Regulation of Mammalian Sperm. Antioxidants, 9, 100.

Yeste, M., Rodríguez‐Gil, J. E., & Bonet, S. (2017). Artificial insemination with frozen‐thawed boar sperm. Molecular Reproduction and Development, 84(9), 802-813.

Zhang, X. G., Liu, Q., Wang, L. Q., Yang, G. S., & Hu, J. H. (2016). Effects of glutathione on sperm quality during liquid storage in boars. Animal science journal, 87(10), 1195–1201.

Round 2

Reviewer 2 Report

The authors have answered all the queries and the manuscript can be accepted in the current format.